# Multilingual Neural Machine Translation with Soft Decoupled Encoding

**Xinyi Wang**[1], **Hieu Pham**[1,2], **Philip Arthur**[3], and **Graham Neubig**[1]

[1]Language Technology Institute, Carnegie Mellon University, Pittsburgh, PA 15213, USA
[2]Google Brain, Mountain View, CA 94043, USA
[3]Monash University, Clayton VIC 3800, Australia
{xinyiw1,hyhieu,gneubig}@cs.cmu.edu,philip.arthur@monash.edu

## Abstract

Multilingual training of neural machine translation (NMT) systems has led to impressive accuracy improvements on low-resource languages. However, there are still significant challenges in efficiently learning word representations in the face of paucity of data. In this paper, we propose Soft Decoupled Encoding (SDE), a multilingual lexicon encoding framework specifically designed to share lexical-level information intelligently without requiring heuristic preprocessing such as pre-segmenting the data. SDE represents a word by its spelling through a character encoding, and its semantic meaning through a latent embedding space shared by all languages. Experiments on a standard dataset of four low-resource languages show consistent improvements over strong multilingual NMT baselines, with gains of up to 2 BLEU on one of the tested languages, achieving the new state-of-the-art on all four language pairs[1].

## 1 Introduction

Multilingual Neural Machine Translation (NMT) has shown great potential both in creating parameter-efficient MT systems for many languages (Johnson et al., 2016), and in improving translation quality of low-resource languages (Zoph et al., 2016; Firat et al., 2016; Gu et al., 2018; Neubig & Hu, 2018; Nguyen & Chiang, 2018). Despite the success of multilingual NMT, it remains a research question how to represent the words from multiple languages in a way that is both parameter efficient and conducive to cross-lingual generalization. The standard sequence-to-sequence (seq2seq) NMT model (Sutskever et al., 2014) represents each lexical unit by a vector from a look-up table, making it difficult to share across different languages with limited lexicon overlap. This problem is particularly salient when translating low-resource languages, where there is not sufficient data to fully train the word embeddings.

Several methods have been proposed to alleviate this data sparsity problem in multilingual lexical representation. The current de-facto standard method is to use subword units (Sennrich et al., 2016; Kudo, 2018), which split up longer words into shorter subwords to allow for generalization across morphological variants or compounds (e.g. "un/decide/d" or "sub/word"). These can be applied to the concatenated multilingual data, producing a shared vocabulary for different languages, resulting in sharing some but not all subwords of similarly spelled words (such as "traducción" in Spanish and "tradução" in Portuguese, which share the root "tradu-"). However, subword-based preprocessing can produce sub-optimal segmentations for multilingual data, with semantically identical and similarly spelled languages being split into different granularities (e.g. "traducción" and "tradu/ção") leading to disconnect in the resulting representations. This problem is especially salient when the high-resource language dominates the training data (see empirical results in Section 4).

In this paper, we propose Soft Decoupled Encoding (SDE), a multilingual lexicon representation framework that obviates the need for segmentation by representing words on a full-word level, but can nonetheless share parameters intelligently, aiding generalization. Specifically, SDE *softly* decouples the traditional word embedding into two interacting components: one component represents

---

[1]The source code is available at https://github.com/cindyxinyiwang/SDE

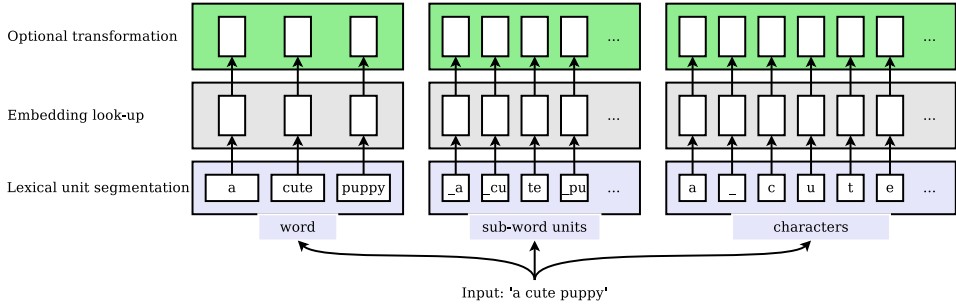

Figure 1: Three steps to compute the lexical representations for the input sequence "a cute puppy", using various lexical unit segmentations.

how the word is spelled, and the other component represents the word's latent meaning, which is shared over all languages present at training time. We can view this representation as a decomposition of language-specific realization of the word's form (i.e. its spelling) and its language-agnostic semantic function. More importantly, our decoupling is done in a *soft* manner to preserve the interaction between these two components.

SDE has three key components: 1) an encoding of a word using character $n$-grams (Wieting et al., 2016); 2) a language specific transform for the character encoding; 3) a latent word embedding constructed by using the character encoding to attend to a shared word embedding space, inspired by Gu et al. (2018). Our method can enhance lexical-level transfer through the shared latent word embedding while preserving the model's capacity to learn specific features for each language. Moreover, it eliminates unknown words without any external preprocessing step such as subword segmentation.

We test SDE on four low-resource languages from a multilingual TED corpus (Qi et al., 2018). Our method shows consistent improvements over multilingual NMT baselines for all four languages, and importantly outperforms previous methods for multilingual NMT that allow for more intelligent parameter sharing but do not use a two-step process of character-level representation and latent meaning representation (Gu et al., 2018). Our method outperforms the best baseline by about 2 BLEU for one of the low-resource languages, achieving new state-of-the-art results on all four language pairs compared to strong multi-lingually trained and adapted baselines (Neubig & Hu, 2018).

## 2 Lexical Representation for Multilingual NMT

In this section, we first revisit the 3-step process of computing lexical representations for multilingual NMT, which is illustrated in Figure 1. Then, we discuss various design choices for each step, as well as the desiderata of an ideal lexical representation for multilingual NMT.

### 2.1 Lexical Unit Segmentation

The first step to compute the neural representation for a sentence is to segment the sentence into lexical units. There are three popular options with different granularities:

- **Word-based** method splits an input sequence into words, often based on white spaces or punctuation. This is perhaps the natural choice for lexical unit segmentation. Early work in NMT all employ this method (Sutskever et al., 2014; Bahdanau et al., 2015).

- **Character-based** method splits an input sequence into characters (Lee et al., 2017).

- **Subword-based** method splits each word into pieces from a small vocabulary of frequently occurring patterns (Sennrich et al., 2016; Kudo, 2018).

## 2.2 EMBEDDING LOOK-UP

After a sentence is segmented into lexical units, NMT models generally look up embeddings from a dictionary to turn each lexical unit into a high-dimensional vector. In the context of multilingual NMT, the lexical unit segmentation method affects this dictionary in different ways.

In **word-based** segmentation, since the number of unique words for each language is unbounded, while computer memory is not, previous work, e.g. Sutskever et al. (2014), resorts to a fixed-size vocabulary of the most frequent words, while mapping out-of-vocabulary words to an ⟨unk⟩ token. For multilingual NMT settings, where multiple languages are processed, the number of words mapped to ⟨unk⟩ significantly increases. Moreover, different languages, even related languages, have very few words that have exactly the same spelling, which leads to the same concept being represented by multiple and independent parameters. This disadvantage hurts the translation model's ability to learn the same concept in multiple languages.

Meanwhile, **character-based** segmentation can effectively reduce the vocabulary size, while maximizing the potential for parameter sharing between languages with identical or similar character sets. However, character segmentation is based on the strong assumption that neural networks can infer meaningful semantic boundaries and compose characters into meaningful words. This puts a large amount of pressure on neural models, requiring larger model sizes and training data. Additionally, training character-based NMT systems is often slow, due to the longer character sequences (Cherry et al., 2018).

**Subword-based** segmentation is a middle ground between word and character segmentation. However, in multilingual translation, the subword segmentation can be sub-optimal, as the subwords from high-resource languages, i.e. languages with more training data, might dominate the subword vocabulary, so that the words in low-resource language can be split into extremely small pieces.

Therefore, existing methods for lexical unit segmentation lead to difficulties in building an effective embedding look-up strategy for multilingual NMT.

## 2.3 OPTIONAL ENCODING TRANSFORMATIONS

Most commonly, the embedding vectors looked up from the embedding table are used as the final lexical representation. However, it is also possible to have multiple versions of embeddings for a single lexicon and combine them through operations such as attentional weighted sum (Gu et al., 2018). Without loss of generality, we can assume there is always a transformation applied to the embedding vectors, and models that do not use such a transformation can be treated as using the identity transformation.

## 2.4 DESIDERATA

To efficiently utilize parameters for multilingual NMT, the lexical representation should have two properties. First, for maximal accuracy, the lexical representation should be able to *accurately represent words in all of the languages under consideration*. Second, for better cross-lingual learning and generalization, such a representation should *maximize the sharing of parameters across languages*.

These two conflicting objectives are difficult to achieve through existing methods. The most common method of using lookup embeddings can only share information through lexical units that overlap between the languages. Subword segmentation strikes a middle ground, but has many potential problems for multilingual NMT, as already discussed in Section 2.2. Although Gu et al. (2018)'s method of latent encoding increases lexical level parameter sharing, it still relies on subwords as its fundamental units, and thus inherits the previously stated problems of sub-word segmentation. We also find in experiments in Section 4 that it is actually less robust than simple lookup when large monolingual data to pre-train embeddings is not available, which is the case for many low-resourced languages.

Next, in Section 3, we propose a novel lexical representation strategy that achieves both desiderata.

## 3 SOFT DECOUPLED ENCODING

Given the conflict between sharing lexical features and preserving language specific properties, we propose SDE, a general framework to represent lexical units for multilingual NMT. Specifically, following the linguistic concept of the "arbitrariness of the sign" (Chandler, 2007), SDE decomposes the modeling of each word into two stages: (1) modeling the language-specific spelling of the word, and (2) modeling the language-agnostic semantics of the word. This decomposition is based on the need for distinguished treatments between a word's semantics and its spelling.

**Semantic representation** is language-agnostic. For example, the English "hello" and the French "bonjour" deliver the same greeting message, which is invariant with respect to the language. SDE shares such semantic representations among languages by querying a list of shared concepts, which are loosely related to the linguistic concept of "sememes" (Greimas, 1983). This design is implemented using an attention mechanism, where the query is the lexical unit representation, and the keys and the values come from an embedding matrix shared among all languages.

Meanwhile, the **word spellings** are more sophisticated. Here, we identify two important observations about word spellings. First, words in related languages can have similar spellings, e.g. the English word "color" and the French word "coleur". In order to effectively share parameters among languages, a word spelling model should utilize this fact. Second, and not contradicting the first point, related languages can also exhibit consistent spelling shifts. For instance, "Christopher", a common name in English, has the spelling "Kryštof" in Czech. This necessitates a learnable rule to convert the spelling representations between such pairs of words in related languages. To account for both points, we use a language-specific transformation on top of a first encoding layer based on character $n$-grams.

### 3.1 EXISTING METHODS

Before we describe our specific architecture in detail (Section 3.2), given these desiderata discussed above, we summarize the designs of several existing methods for lexical representation and our proposed SDE framework in Table 1. Without a preprocessing step of subword segmentation, SDE can capture the lexical similarities of two

| Method | Lex Unit | Embedding | Encoding |
|---|---|---|---|
| Johnson et al. (2016) | Subword | joint-Lookup | Identity |
| Lee et al. (2017) | Character | joint-Lookup | Identity |
| Gu et al. (2018) | Subword | pretrain-Lookup | joint-Lookup + Latent |
| Ataman & Federico (2018) | Word | character $n$-gram | Identity |
| SDE | Word | character $n$-gram | Identity + Latent |

Table 1: Methods for lexical representation in multilingual NMT. *joint-Lookup* means the lookup table is jointly trained with the whole model. *pretrain-Lookup* means the lookup table is trained independently on monolingual data.

related languages through the character $n$-gram embedding while preserving the semantic meaning of lexicons through a shared latent embedding.

### 3.2 DETAILS OF SOFT DECOUPLED ENCODING

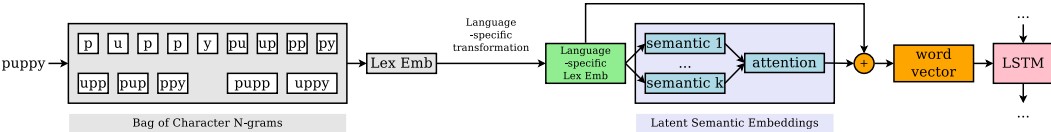

Figure 2: SDE computes the embedding for the word "puppy". Both character $n$-grams embeddings and latent semantic embeddings are shared among all languages.

As demonstrated in Figure 2, given a word $w$ in a multilingual corpus from language $L_i$, SDE constructs the embedding of $w$ in three phases.

**Lexical Embedding.** We maintain an embedding matrix $\mathbf{W}_c \in \mathbb{R}^{C \times D}$, where $D$ is the embedding dimension and $C$ is the number of $n$-grams in our vocabulary. Out-of-vocab $n$-grams are mapped

to a designated token $\langle unk \rangle$. $\mathbf{W}_c$ is shared among all languages. Following Wieting et al. (2016), for each word $w$, we first compute the bag of character $n$-grams of $w$, denoted by BoN($w$), which is a sparse vector whose coordinates are the appearance counts of each character $n$-gram in $w$. For instance, the characters $n$-grams with $n = 1, 2, 3, 4$ of the word "puppy" are shown in Figure 2. We then look up and add the rows of $\mathbf{W}_c$ according to their corresponding counts, and apply a $\tanh$ activation function on the result

$$c(w) = \tanh(\text{BoN}(w) \cdot \mathbf{W}_c). \tag{1}$$

**Language-specific Transformation.** To account for spelling shifts between languages (c.f. Section 3), we apply an language-specific transformation to normalize away these differences. We use a simple fully-connected layer for this transformation. In particular, for language $L_i$, we have

$$c_i(w) = \tanh(c(w) \cdot \mathbf{W}_{L_i}), \tag{2}$$

where $\mathbf{W}_{L_i} \in \mathbb{R}^{D \times D}$ is the transformation matrix specific to language $L_i$.

**Latent Semantic Embedding.** Finally, to model the shared semantics of words among languages, we employ an embedding matrix $\mathbf{W}_s \in \mathbb{R}^{S \times D}$, where $S$ is the number of core semantic concepts we assume a language can express. Similar to the lexical embedding matrix $\mathbf{W}_c$, the semantic embedding matrix $\mathbf{W}_s$ is also shared among all languages.

For each word $w$, its language-specific embedding $c_i(w)$ is passed as a query for an attention mechanism (Luong et al., 2015) to compute a weighted sum over the latent embeddings

$$e_{\text{latent}}(w) = \text{Softmax}(c_i(w) \cdot \mathbf{W}_s^\top) \cdot \mathbf{W}_s. \tag{3}$$

Finally, to ease the optimization of our model, we follow Vaswani et al. (2017) and add the residual connection from $c_i(w)$ into $e_{\text{latent}}(w)$, forming the Soft Decoupled Encoding embedding of $w$

$$e_{\text{SDE}}(w) = e_{\text{latent}}(w) + c_i(w). \tag{4}$$

## 4 EXPERIMENT

We build upon a standard seq2seq NMT model for all experiments. Except for the experiments in Section 4.6, we run each experiment with 3 different random seeds, and conduct significance tests for the results using the paired bootstrap (Clark et al., 2011).

### 4.1 DATASETS

We use the 58-language-to-English TED corpus for experiments. Following the settings of prior works on multilingual NMT (Neubig & Hu, 2018; Qi et al., 2018), we use three low-resource language datasets: Azerbaijani (aze), Belarusian (bel), Galician (glg) to English, and a slightly higher-resource dataset, namely Slovak (slk) to English. Each low-resource language is paired with a related

| LRL | Train | Dev | Test | HRL | Train |
|-----|-------|-----|------|-----|-------|
| aze | 5.94k | 671 | 903 | tur | 182k |
| bel | 4.51k | 248 | 664 | rus | 208k |
| glg | 10.0k | 682 | 1007 | por | 185k |
| slk | 61.5k | 2271 | 2445 | ces | 103k |

Table 2: Statistics of our datasets. LRL and HRL mean Low-Resource and High-Resource Language.

high-resource language: Turkish (tur), Russian (rus), Portuguese (por), and Czech (ces) respectively. Table 2 shows the statistics of each dataset.

### 4.2 BASELINES

For the baseline, we use the standard lookup embeddings for three granularities of lexical units: (1) word: with a fixed word vocabulary size of 64,000 for the concatenated bilingual data; (2) sub-joint: with BPE of 64,000 merge operations on the concatenated bilingual data; and (3) sub-sep: with BPE separately on both languages, each with 32,000 merge operations, effectively creating a vocabulary of size 64,000. We use all three settings to compare their performances and to build a competitive baselines. We also implement the latent embedding method of Gu et al. (2018). We use 32,000 character $n$-gram with $n = \{1, 2, 3, 4, 5\}$ from each language and a latent embedding size of 10,000.

### 4.3 RESULTS

Table 3 presents the results of SDEand of other baselines. For the three baselines using lookup, sub-sep achieves the best performance for three of the four languages. Sub-joint is worse than sub-sep although it allows complete sharing of lexical units between languages, probably because sub-joint leads to over-segmentation for the low-resource language. Our reimplementation of universal encoder (Gu et al., 2018) does not perform well either, probably because the monolingual embedding is not trained on enough data, or the hyperparamters for their method are harder to tune. Meanwhile, SDE outperforms the best baselines for all four languages, without using subword units or extra monolingual data.

| Lex Unit | Model | aze | bel | glg | slk |
|---|---|---|---|---|---|
| Word | Lookup | 7.66 | 13.03 | 28.65 | 25.24 |
| Sub-joint | Lookup | 9.40 | 11.72 | 22.67 | 24.97 |
| Sub-sep | Lookup (Neubig & Hu, 2018)[2] | 10.90 | 16.17 | 28.10 | 28.50 |
| Sub-sep | UniEnc (Gu et al., 2018)[3] | 4.80 | 8.13 | 14.58 | 12.09 |
| Word | SDE | 11.82* | 18.71* | 30.30* | 28.77† |

Table 3: BLEU scores on four language pairs. Statistical significance is indicated with $*$ ($p < 0.0001$) and † ($p < 0.05$), compared with the best baseline.

| Model | aze | bel | glg | slk |
|---|---|---|---|---|
| SDE | 11.82 | 18.71 | 30.30 | 28.77 |
| -Language Specific Transform | 12.89* | 18.13† | 30.07 | 29.16† |
| -Latent Semantic Embedding | 7.77* | 15.66* | 29.25* | 28.15* |
| -Lexical Embedding | 4.57* | 8.03* | 13.77* | 7.08* |

Table 4: BLEU scores after removing each component from SDE-com. Statistical significance is indicated with $*$ ($p < 0.0001$) and † ($p < 0.005$), compared with the full model in the first row.

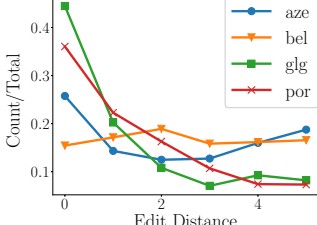

Figure 3: Percentage of words by the edit distance from the matching words in the high-resource language.

### 4.4 ABLATION STUDIES

We next ablate various features of SDE by removing each of the three key components, and show the results in Table 4. Removing the latent semantic embedding and lexical embedding consistently harms the performance of the model. The effect of the language specific transformation is smaller and is language dependent. Removing the language specific transformation does not lead to significant difference for glg. However, it leads to a 0.8 gain in BLEU for aze, a 0.3 gain for slk, but about a 0.6 decrease for bel. A further inspection of the four language pairs shows that the language specific transform is more helpful for training on languages with fewer words of the same spelling. To quantify this, we extract bilingual dictionaries of the low-resource languages and their paired high-resource languages, and measure the edit distance between the character strings of the aligned words. We use FastAlign (Dyer et al., 2013) to extract aligned words between the source language and English from the parallel training data, then match the English side of two related languages

---

[2]For all tasks in our experiments, our reimplementation of Neubig & Hu (2018) achieves similar or slightly higher BLEU scores than originally reported (aze: 10.9; bel: 15.8; glg: 27.3; slk: 25.5). We suspect the difference is because we use a different tokenizer from Neubig & Hu (2018). Details are in our open-sourced software.

[3]To ensure the fairness of comparison with other methods, we only train the monolingual embedding from the parallel training data, while Gu et al. (2018) used extra monolingual data from the Wikipedia dump. We have also tried testing our reimplementation of their method with trained monolingual embedding from the Wikipedia dump, but achieved similar performance.

to get their dictionary. Figure 3 shows the percentage of the word pairs grouped by their edit distance. Among the four languages, bel has the lowest percentage of words that are exactly the same with their corresponding words in the high-resource language (0 edit distance), indicating that the language specific transform is most important for divergent languages.

## 4.5 EFFECT OF SUBWORDS

Although using subwords in NMT can eliminate unknown words and control the vocabulary size, it might not be optimal for languages with rich morphology (Ataman & Federico, 2018). We also show in Table 3 that certain choices when using subwords on multilingual data (e.g. whether to train the segmentation on the concatenated multilingual data or separately) have a large effect on the performance of the model. SDE achieves superior performance without using subwords, so it can avoid the risk of sub-optimal segmentation. Still, we test SDE with subwords to study its effect on our framework. The results are shown in Table 5. We test two methods of using subwords: 1)

| Lex Unit | Model | aze | bel | glg | slk |
|----------|---------|------|------|------|------|
| Word | SDE | 11.82 | 18.71 | 30.30 | 28.77 |
| Sub-sep | SDE | $12.37^{\dagger}$ | $16.29^{*}$ | $28.94^{*}$ | $28.35^{\dagger}$ |
| Word | SDE-sub | 12.03 | $18.16^{\dagger}$ | $31.16^{*}$ | 28.86 |

Table 5: BLEU scores on four language pairs. Statistical significance is indicated with $*$ ($p < 0.0001$) and $\dagger$ ($p < 0.005$), compared with the setting in row 1.

we use sub-sep as lexical units, and encode its lexical representation using the character $n$-grams of the subwords; 2) we use words as lexical units, but use its subword pieces instead of the character $n$-grams to construct its lexical representation. We use SDE-sub to indicate the second way of using subwords. When using SDE with sub-sep as lexical units, the performance slightly increases for aze, but decreases for the other three language. Therefore, it is generally better to directly use words as lexical units for SDE. When we use words as lexical units but replace character $n$-gram with subwords, the performance on two of the languages doesn't change significantly, while the performance decreases for bel and increases for glg.

We also examine the performance of SDE and the best baseline sub-sep with different vocabulary size and found that SDE is also competitive with a small character $n$-gram vocabulary of size 8K. Details can be found in Appendix A.2.

## 4.6 TRAINING ON ALL LANGUAGES

To further compare SDE's ability to generalize to different languages, we train both SDE and sub-sep on the low-resource languages paired with all four high-resource languages. The results are listed in Table 6. For bel, SDE trained on all languages is able to improve over just training with bilingual data by around 0.6 BLEU. This is the best result on bel, with around 3 BLEU over the best baseline. The performance of sub-sep, on the other hand, decreases by around 1.5 BLEU when training on all languages for bel. The performance of both methods de-

| Lex Unit | Model | aze | | bel | |
|----------|--------|-------|-------|-------|-------|
| | | bi | all | bi | all |
| Sub-sep | Lookup | 11.25 | 8.10 | 16.53 | 15.16 |
| Word | SDE | 12.25 | 12.09 | 19.08 | 19.69 |

Table 6: BLEU scores for training with all four high-resource languages.

creases for aze when using all languages. SDE only slightly loses 0.1 BLEU while sub-sep loses over 3 BLEU.

## 4.7 WHY DOES SDE WORK BETTER?

The SDE framework outperforms the strong sub-sep baseline because it avoids sub-optimal segmentation of the multilingual data. We further inspect the improvements by calculating the word F-measure of the translated target words based on two properties of their corresponding source

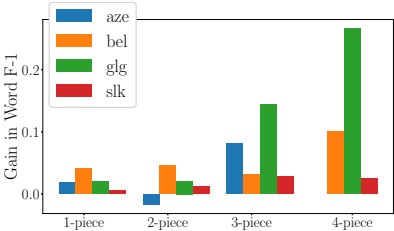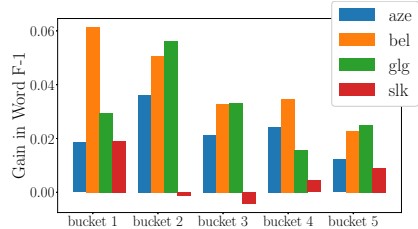

Figure 4: Gain in word F-measure of SDE over sub-sep. *Left*: the target words are bucketed by the number of subword pieces that their corresponding source words are segmented into. *Right*: the target words are bucketed by the edit distance between their source words and the corresponding words in the high resource language.

words: 1) the number of subwords they were split into; 2) the edit distance between their corresponding words in the related high-resource language. From Figure 4 *left*, we can see that SDE is better at predicting words that were segmented into a large number of subwords. Figure 4 *right* shows that the gain peaks on the second bucket for 2 languages and the first bucket for bel and slk, which implies that SDE shows more improvements for words with small but non-zero edit distance from the high-resource language. This is intuitive: words similar in spelling but with a few different characters can be split into very different subword segments, while SDE can leverage the lexical similarities of word pairs with slightly different spelling.

## 4.8 QUALITATIVE ANALYSIS

Table 7 lists a few translations for both sub-sep and SDE. We can see that SDE is better at capturing functional words like "if" and "would". Moreover, it translates "climatologist" to a related word "weather", probably from the prefix "climat" in glg, while sub-sep gives the totally unrelated translation of "college friend". Some examples of bilingual lexicons that can be better captured by SDE can be found in Appendix A.3.

| glg | eng | sub-sep | SDE |
|---|---|---|---|
| Pero non temos a tecnoloxía para resolver iso, temos? | But we don't have a technology to solve that, right? | But we don't have the technology to solve that , we have? | But we don't have the technology to solve that, do we? |
| Se queres saber sobre o clima, preguntas a un climatólogo. | If you want to know about climate, you ask a climatologist. | If you want to know about climate, you're asking a college friend. | If you want to know about climate, they ask for a weather. |
| Non é dicir que si tivesemos todo o diñeiro do mundo, non o quereríamos facer. | It's not to say that if we had all the money in the world, we wouldn't want to do it . | It's not to say that we had all the money in the world, we didn't want to do it . | It's not to say that if we had all the money in the world, we wouldn't want to do it. |

Table 7: Examples of glg to eng translations.

## 5 RELATED WORKS

In multilingual NMT, several approaches have been proposed to enhance parameter sharing of lexical representations. Zoph et al. (2016) randomly assigns embedding of a pretrained NMT model to the vocabulary of the language to adapt, which shows improvements over retraining the new embeddings from scratch. Nguyen & Chiang (2018) propose to match the embedding of the word piece that overlaps with the vocabulary of the new language. When training directly on concatenated data, it is also common to have a shared vocabulary of multilingual data (Neubig & Hu, 2018; Qi et al., 2018). Gu et al. (2018) propose to enhance parameter sharing in lexical representation by a latent embedding space shared by all languages.

Several prior works have utilized character level embeddings for machine translation (Cherry et al., 2018; Lee et al., 2017; Ataman & Federico, 2018), language modeling (Kim et al., 2016; Józefowicz

et al., 2016), and semantic parsing (Yih et al., 2014). Specifically for NMT, fully character-level NMT can effectively reduce the vocabulary size while showing improvements for mulitlingual NMT (Lee et al., 2017), but it often requires much longer to train (Cherry et al., 2018). Ataman & Federico (2018) shows that character $n$-gram encoding of words can improve over BPE for morphologically rich languages.

## 6  CONCLUSION

Existing methods of lexical representation for multilingual NMT hinder parameter sharing between words that share similar surface forms and/or semantic meanings. We show that SDE can intelligently leverage the word similarities between two related languages by softly decoupling the lexical and semantic representations of the words. Our method, used without any subword segmentation, shows significant improvements over the strong multilingual NMT baseline on all languages tested.

**Acknowledgements:** The authors thank David Mortensen for helpful comments, and Amazon for providing GPU credits. This material is based upon work supported in part by the Defense Advanced Research Projects Agency Information Innovation Office (I2O) Low Resource Languages for Emergent Incidents (LORELEI) program under Contract No. HR0011-15-C0114. The views and conclusions contained in this document are those of the authors and should not be interpreted as representing the official policies, either expressed or implied, of the U.S. Government. The U.S. Government is authorized to reproduce and distribute reprints for Government purposes notwithstanding any copyright notation here on.

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

# A APPENDIX

## A.1 TRAINING DETAILS

- We use a 1-layer long-short-term-memory (LSTM) network with a hidden dimension of 512 for both the encoder and the decoder.
- The word embedding dimension is kept at 128, and all other layer dimensions are set to 512.
- We use a dropout rate of 0.3 for the word embedding and the output vector before the decoder softmax layer.
- The batch size is set to be 1500 words. We evaluate by development set BLEU score for every 2500 training batches.
- For training, we use the Adam optimizer with a learning rate of 0.001. We use learning rate decay of 0.8, and stop training if the model performance on development set doesn't improve for 5 evaluation steps.

## A.2 EFFECT OF VOCABULARY SIZE

We examine the performance of SDE and the best baseline sub-sep, with a character $n$-gram vocabulary and sub-word vocabulary respectively, of size of 8K, 16K, and 32K. We use character $n$-gram of $n = \{1, 2, 3, 4\}$ for 8K and 16K vocabularies, and $n = \{1, 2, 3, 4, 5\}$ for the 32K vocabulary. Figure 5 shows that for all four languages, SDE outperforms sub-sep with all three vocabulary sizes. This shows that SDE is also competitive with a relatively small character $n$-gram vocabulary.

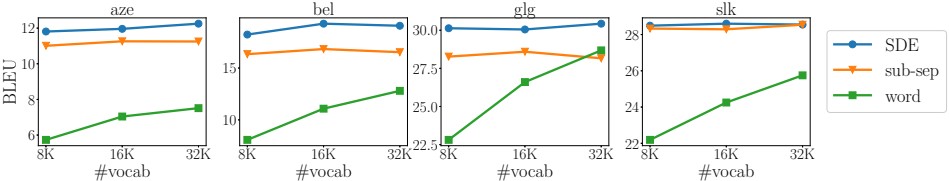

Figure 5: Performance on three different vocabulary size (results of a single random seed).

## A.3 EXAMPLE BILINGUAL WORDS

Table 8 lists some words and their subwords from bel and its related language rus. We can see that subwords fail to capture all the lexical similarities between these words, and sometimes the word pairs are segmented into different number of pieces.

| bel | | rus | | eng |
|---|---|---|---|---|
| word | subword | word | subword | |
| фінансавыя | фінансавы я | финансовых | финансовы х | financial |
| стадыён | стады ён | стадион | стадион | stadium |
| розных | розны х | разных | разны х | different |
| паказаць | паказа ць | показать | показать | show |

Table 8: Bilingual word pairs and their subword pieces.

## A.4 ANALYSIS OF ATTENTION OVER LATENT EMBEDDING SPACE

In this section we compare the attention distribution over the latent embedding space of related languages, with the intuition that words that mean the same thing should have similar attention distributions. We calculate the KL divergence of the attention distribution for word pairs in both the LRL and HRL. Figure 6 shows that the lowest KL divergence is generally on the diagonals representing words with identical meanings, which indicates that similar words from two related languages

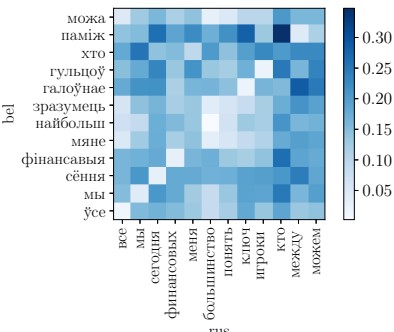 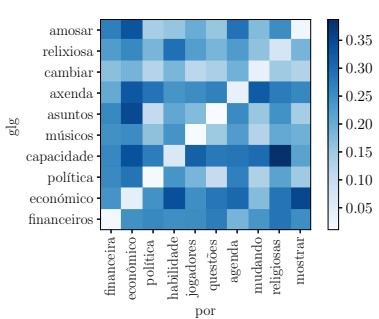

Figure 6: KL divergence of attention over latent embedding space between words from two related languages. Word pairs that match at the diagonal have similar meanings. *Left*: bel-rus. *Right*: glg-por.

| glg | eng | por | eng |
|---------|-------|---------|--------------|
| cando | when | quando | when |
| | | caindo | failing down |
| etiqueta | label | rótulo | label |
| | | riqueza | wealth |

Table 9: Words in glg-por that have the same meaning but different spelling, or similar spelling but different meaning.

tend to have similar attention over the latent embedding space. Note that this is the case even for words with different spellings. For example, Figure 6 *right* shows that the KL divergence between the glg word "músicos" (meaning "musicians"), and the por word of the closet meaning among the words shown here, "jogadores" (meaning "players"), is the smallest, although their spellings are quite different.

## A.5 ANALYSIS OF WORD VECTORS FROM SDE

In this section, we qualitatively examine the location of word vectors at different stages of SDE. We reduce the word vectors to two dimensions using t-SNE (van der Maaten & Hinton, 2008). In particular, we focus on two groups of words shown in Table 9 from glg-por, where each word from glg is paired with two words from por, one with the same meaning but different spelling, while the other has similar spelling but different meaning. Figure 7 *left* shows the embeddings derived from the character *n*-grams. At this stage, the word "cando" is closer to "caindo", which has a different meaning, than "quando", which has the same meaning but a slightly more different spelling. The word "etiqueta" lies in the middle of "rótulo" and "riqueza". After the whole encoding process, the location of the words are shown in Figure 7 *right*. At the final stage, the word "cando" moves closer to "quando", which has the same meaning, than "caindo", which is more similar in spelling. The word "etiqueta" is also much closer to "rótulo", the word with similar meaning, and grows further apart from "riqueza".

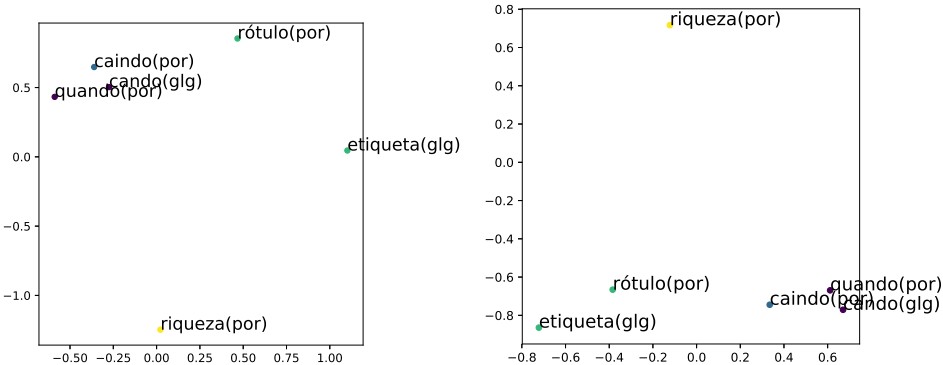

Figure 7: T-SNE visualizations of the embeddings of words in Table 9 encoded after the character *n*-gram embedding stage (*Left*), or after the full process of SDE(*Right*). Words of the same color have similar meanings, and the language code of the word is placed in the parenthesis. It can be seen that the character embedding stage is more sensitive to lexical similarity, while the full SDEmodel is more sensitive to similarity in meaning.

