# OpenReview forum: "Multilingual Neural Machine Translation With Soft Decoupled Encoding"
_ICLR.cc/2019/Conference_

### Official Review · AnonReviewer1 · 2018-11-02
**An interesting word representation model with good ablation experiments**

**Rating:** 7
**Confidence:** 4

**Review:**

This paper presents an approach to creating word representations that operate at both the sub-word level and generalise across languages. The paper presents soft decoupled encoding as a method to learn word representations from weighted bags of character-n grams, a language specific transformation layer, and a "latent semantic embedding" layer. The experiments are conducted over low-resource languages from the multilingual TED corpus. The experiments show consistent improvements compared to existing approaches to training translation models with sub-word representations. The ablation studies in Section 4.4 are informative about the relative importance of different parts of the proposed model.

Can you comment on how your model is related to the character-level CNN of Lee et al. (TACL 2017)?

In the experiments, do you co-train the LRLs with the HRLs? This wasn't completely clear to me from the paper. In Section 4.2 you use phrases like "concatenated bilingual data" but I couldn't find an explicit statement that you were co-training on both language pairs.

What does it mean for the latent embedding to have a size of 10,000? Does that mean that W_s is a 10,000 x D matrix?

Is Eq (4) actually a residual connection, as per He et al. (CVPR 2016)? It looks more like a skip connection to me.

Why do you not present results for all languages in Section 4.6?

What is the total number of parameters in the SDE section of the encoder? The paper states that you encode 1--5 character n-grams, and presumably the larger the value of N, the sparser the data, and the larger the number of parameters that you need to estimate.

For which other tasks do you think this model would be useful?

---

> ### Author Response · Authors · 2018-11-08
> **Response to the reviewer comments**
>
> We thank the reviewer for the comments and insightful questions:
>
> 1. Comparison to Lee et al.: SDE share some of the motivations with Lee et. al.: 1) both models don’t require segmenting into subwords; 2) both methods can leverage the similarity in spelling of words from different languages. SDE also has several advantages over Lee et. al. : 1) SDE still use words as lexical units, while Lee et. al. use characters. Therefore, the fully character-based NMT model in Lee et. al. needs to operate on much longer sequence than SDE, potentially making it much slower than word-level NMT; 2) The model in Lee et. al. needs a large number of other hyperparameters, like the number of kernels, kernel size, size of max pooling stride, while for SDE we only need to decide the size of the latent embedding space. While we did not directly compare in this paper, we have previously found that it is quite difficult to get good results with fully character-based models due to this necessity of careful hyper-parameter tuning and long experimental time. We can try to add a comparison if the reviewer thinks it would be informative, but we likely won’t be able to do so by the end of the rebuttal period.
>
> 2. Joint training?: Yes we do co-train the LRL with the HRL. We will clarify this in the final version.
>
> 3. Latent embedding size?: Yes the latent embedding size of 10,000 means the W_s matrix is 10,000*D
>
> 4. Residual connection?: In equation 4, c_i(w) can be seen as the input x, and e_latent(w) is a function that takes in input x. e_SDE(w) is the sum of the function of x and the input x, which we believe follows the definition of residual connection in He et. al.
>
> 5. Why not other languages?: To use all HRL data, we need to co-train the LRL with all of the HRL data, so for each LRL, the time it takes to converge is much longer. Therefore we only presented two experiments here to show that SDE has more desirable performance than the sep-sub baseline when using data from languages that are less related to the LRL. We can probably add these to the final version of the paper, however, if we start experiments now.
>
> 6. Number of parameters: For the experiments in the paper, SDE NMT model has around 22M parameters, and the sub-sep NMT model has 18M parameters. The extra parameters are from the character n-gram embedding. Moreover, we show in Appendix A2 that  SDE still outperforms the sub-sep baseline when using a smaller character n-gram vocabulary.
>
> 7.Other tasks: SDE is a model-agnostic word encoding framework. First, it can easily integrate with other neural machine translation models, such as the Transformer model. Second, it can be used together with any neural models that need to encode language data. For example, we can use SDE to encode multilingual data for named entity recognition or other natural language analysis tasks for low-resource languages.

---

### Official Review · AnonReviewer2 · 2018-11-02
**Well-motivated problem and reasonable solution. Desire a few more experiments and clarifications.**

**Rating:** 6
**Confidence:** 4

**Review:**

This paper focuses on the problem of word representations in multilingual NMT system. The idea of multilingual NMT is to share data among multiple language pairs. Crucially this requires some way to tie the parameters of words from different languages, and one popular method is to share subword units among languages. The problem is that subword units in different languages may not be semantically equivalent, and many semantically-equivalent concepts are not represented by the same subwords. This paper proposes an alternative way to share word representation, in particular by proposing a common set of "semantic" concept vectors across languages which are then folded into the word representations via attention.

The problem is well-motivated and the proposed solution is reasonable. Previous works such as (Gu et. al. 2018) have been motivated in a similar fashion, and the proposed solution seems to outperform it on the TED dataset of Qi et. al. 2018.

The experiments are informative. The main open questions I have are:

(a) Varying the latent embedding size. It seems like only 10,000 is tried. Since this is the main contribution of the work, it will be desirable to see results for different sizes. Is the method sensitive to this hyperparameter? Also suggestions on how to pick the right number based on vocabulary size, sentence size, or other language/corpus characteristics will be helpful.

(b) What do the latent embeddings look like? Intuitively will they be very different from those from Gu et. al. 2018 because you are using words rather than subwords as the lexical unit?

(c) The explanation for why your model outperforms Gu et. al. 2018 seems insufficient -- it would be helpful to provide more empirical evidence in the ablation studies in order really understand why your method, which is similar to some extent, is so much better.

The paper is generally clear. Here are few suggestions for improvement:

- Table 1: Please explain lex unit, embedding, encoding in detail. For example, it is not clear what is joint-Lookup vs. pretrain-Lookup. It can be inferred if one knows the previous works, but to be self-contained, I would recommend moving this table and section to Related Works and explaining the differences more exactly.

- Sec 4.2: Explain the motivation for examining the three different lexical units.

- Table 3: "Model = Lookup (ours)" was confusing. Do you mean "our implementation of Neubig & Hu 2018? Or ours=SDE? I think the former?

- Are the word representions in Eq 4 defined for each word type or word token? In other words, for the same word "puppy" in two different sentences in the training data, do they have the same attention and thus the same e_SDE(w)? You do not have different attentions depending on the sentence, correct? I think so, but please clarify. (Actually, Figure 2 has a LSTM which implies a sentential context, so this was what caused the potential confusion).

- There are some inconsistencies in the terms: e.g. latent semantic embedding vs latent word embedding. Lexical embedding vs Character embedding. This makes it a bit harder to line up Sec 4.4 results with Sec 3.2 methods.

- Minor spelling mistakes. e.g. dependant -> dependent. Please double-check for others.

---

> ### Author Response · Authors · 2018-11-08
> **Response to the reviewer comments**
>
> We thank the reviewer for the insightful questions and suggestions. We have performed additional experiments and will add more experimental results and revise the final version. Here are some partial response to address the questions:
>
> (a) Varying the embedding size: In our first few experimental runs, we varied the latent embedding size of 5000, 10,000, and 15,000 for the bel-rus dataset. We found that SDE with embedding size of 10000 performs the best, so we just fixed the number for all the rest of the experiments. We started some new experiments varying the latent embedding size of 5000, 10,000, and 15,000 for azetur and belrus dataset, and here are the results on the test set: for azetur, the test BLEU scores are 12.43, 11.66, 10.65 respectively, while the sub-sep baseline is 10.42; for belrus, the test BLEU scores are 18.74, 19.03, 17.76 respectively, while the sub-sep baseline is 17.14. Therefore, SDE out-performs the sub-sep baseline with all three different latent embedding sizes. Moreover, SDE with a relatively small latent embedding size can usually achieve good performance. We will update the paper accordingly.
>
> (b) What do the embeddings look like?: The reviewer raised a reasonable point that our latent embedding might look very different from Gu et al.. This is certainly true because of the following differences between SDE and the encoder in Gu et al.: 1) SDE uses words as lexical units, while Gu et. al. uses word pieces; 2) SDE constructs a character-ngram embedding that can effectively capture the similarity in spelling of words in related languages (see appendix A3), while Gu et. al requires standard pre-trained word embedding; 3)  SDE adds back the lexical character n-gram embedding to the latent semantic embedding to create the final word embedding, while Gu et. al. only uses the latent embedding. Therefore, the latent embedding space in SDE has a very different functionality than the one in Gu et. al. The latent embedding itself is difficult to visualize, mainly because there is not a one-to-one correspondence between the vocabulary and the latent embedding, but we will try to visualize the word vectors at different stages in the SDE encoding process and will update once we have some results.
>
> (c) Explanation for improvement over Gu et al.: We think the gains are from two main differences: 1) SDE has a character n-gram embedding that can capture the character overlap of similar words in different languages, while the standard word embedding in Gu et. al. cannot leverage this information. This is particularly important in the case of languages with similar spelling, like the ones we used in our experiments; 2) SDE has a residual connection that adds the lexical character n-gram embedding back to the latent semantic embedding, while Gu et. al. only has the latent embedding. In Table 4., we show that the performance of SDE is significantly harmed by removing the residual connection of the character n-gram embedding. In fact, the results after removing character n-gram embedding (row 4 in Table 4) are somewhat comparable to the results of Gu et. al. (row 5 in Table 3). Another advantage of SDE is that it can directly use words as lexical units, while the model in Gu et. al. uses subwords. As the ablation experiments in Table 5 suggest, using subwords as lexical units in general is not as good as using words directly with SDE.
>
>
> We are very grateful for the careful comments about the presentation of the paper. We will address these comments in the updated version of the paper:
> 1. In Table 3, (ours) means our implementation of Neubig & Hu 2018.
>
> 2. For the same word “puppy” in two different sentences of the same language, we will have the same e_SDE(w). The LSTM in Figure 2 is meant to represent the encoder RNN. We will make this clear.

---

> > ### Comment · AnonReviewer2 · 2018-11-15
> > **Thanks for the clarification**
> >
> > Thanks for the clarification.
> >
> > (a) Nice! This is good to know. I would also suggesting testing the limits -- making the embedding so small or so big such that results degrade. This will give a better sense of the sensitivity.
> >
> > (b) Visual inspection might be difficult but perhaps you can include some kind of kNN of of the same words/subwords. If it's not possible to find any trends, then it's ok, I understand. I wouldn't cherry pick too much.
> >
> > (c) This is a very nice explanation. Please put it in the paper if you can, so differences with Gu et. al. is clear. By the way, I hadn't realized the residual connections were so important. I think standardizing some of the terminology in different parts of the paper would make it clearer.

---

> > > ### Author Response · Authors · 2018-11-21
> > > **Replies**
> > >
> > > Thank you so much for the suggestions!
> > > a) We will try testing the limits and include some results in the final version.
> > >
> > > b) We tried different clustering methods but it's hard to interpret the results. However, we added two more visualizations, one for attention over latent embedding matrix and the other for word vectors, in the appendix.
> > >
> > > c) Thanks for pointing this out! We will include the clarification in the final version. The current updated version fixed some terminology discrepancies and we hope it's less confusing now.

---

> ### Public Comment · (anonymous) · 2018-12-13
> **More fair comparison to Gu et. al. 2018**
>
> The paper is quite nice, there are a lot of similarity to Gu et. al.  though. One clear difference is that this paper uses char-ngram embedding (wieting 16)  while Gu. et. al.  is using pre-trained embedding. The attention-based latent representation is quite similar in both papers.
>
> Though Gu et. al clearly states that the model  requires monolingual data to train the embedding,  the authors of this paper  did not do so and just trained  the mono-lingual embedding on the tiny parallel data sentences.  It is not hard at all to get sufficient monolingual data to train embedding for any language (i.e. from wikipedia dump) which would give enough data to train  reliable embeddings for any language including the four ones reported here; see Muse for example.
> I find the results in Table 3  to be quite misleading since it lists Gu et. al. for comparison while depriving the model from its essential requirement of monolingual embedding.  I think you should report the result both with and without mono-lingual embedding to have a more fair comparison.
> If the scope of this paper is focusing on not using any monolingual data, then it should not compare with a method that requires monolingual data.
>
> The response above  to one the reviewers about the contrast with Gu et. al. is not accurate as well since it states that Gu. et. al. does not have residual connection, while it realy has, see eqn 8.  The main difference is that this paper has char-ngram reprehension while Gu. et. has pre-trained embedding which is not usable without monolingual data.
>
> The similarities between this work and Gu et. al. are clearly more than the differences;   and it would be more sound to have better analysis in the paper.
>
> I think the authors should consider adding more fair comparison and discussion on this issue.

---

> > ### Author Response · Authors · 2018-12-13
> > **Response**
> >
> > First, thank you for the comments! We’d like to note that we definitely didn't mean to mis-represent or do an unfair comparison, and we apologize if it came across this way. After receiving this comment we do realize that this might not have been clear enough in the paper, so in order to remedy this, we will remove the name Gu et al. from the table and explain more clearly and highlight the differences in the main text.
> >
> > To explain more completely the reason why we didn’t use pre-trained embeddings for this architecture: this was also to preserve fairness, specifically in the comparison with the proposed method. We don’t believe that comparing methods with different resource requirements is fair, and thus we used the same data to train all of the methods tested in our paper. Note that all methods in our paper could also utilize pre-training, but this is complicated because some of our models use subwords, some use words, and some use character embeddings as input, so we would need to appropriately pre-train all of these different embeddings. That being said, this is possible to do and we’ll try to do so in the near future. Unfortunately we won’t be able to do it during the review period, however, as we got this comment at the last minute and most of the authors on the paper are already on winter holidays. We do hope that our response to the other reviewers’ comments has demonstrated that we will make a good-faith effort in this regard, and we expect that our full SDE method will still do significantly better due to the high lexical overlap between related languages.
> >
> > We would also like to qualify the statement that “it is not hard at all to get sufficient monolingual data to train embedding for any language.” While we experimented on moderately resourced languages like Galician and Azerbaijani for the sake of this paper (because these are the languages where well-curated MT datasets are available), we are actually interested in translation of languages with even fewer resources, such as African languages. Despite having millions of speakers, high-quality monolingual text for these languages is hard to come by. For example, Kinyarwanda, the official language of Rwanda has 9.8 million speakers, but only 1,822 articles on Wikipedia. The situation is even worse for languages that have fewer speakers or a less official status.
> >
> > We're sorry about the mis-characterization of the residual connections in the comments, this was a mistake on our part (in the reviewer comment, not the experiments). Equation 8 in Gu et. al. shows that the residual connection is applied only to the top K words, while our method uses residual connection to all words and thus doesn’t require this heuristic. In our current reimplementation of Gu et. al., we do indeed have a residual connection to the top K words. In addition, we also tried adding residual connection to all words for the Gu et. al. method and it seems to hurt the performance.

---

### Official Review · AnonReviewer4 · 2018-11-10
**Interesting and clean idea for multilingual lexicon sharing.**

**Rating:** 6
**Confidence:** 5

**Review:**

Overall:
This paper proposed soft decoupled encoding (SDE), a special multilingual lexicon encoding framework which can share lexical-level information without requiring heuristic preprocessing. Experiments for low-resource languages show consistent improvements over strong multilingual NMT baselines.

General Comments:
To me this paper is very interesting and is nicely summarized and combined previous efforts in two separated directions for sharing multilingual lexicons: based on the surface similarity (how the word is spelled, e.g. subword/char-level models), and based on latent semantic similarity (e.g. Gu et.al. 2018). However, in terms of the proposed architecture, it seems to lack some novelty. Also, more experiments are essential for justification.

I have some questions:
(1) One of the motivation proposed by Gu et.al. 2018 is that spelling based sharing sometimes is difficult/impossible to get (e.g. distinct languages such as French and Korean), but monolingual data is relatively easy to obtain. Some languages such as Chinese is not even “spelling” based. Will distinct languages still fit in the proposed SDE? In my point of view, it will break the “query” vector to attention to the semantic embeddings.
(2) How to decide the number of core semantic concepts (S) in the latent semantic embeddings? Is this matrix jointly trained in multilingual setting?
(3) Is the latent semantic embeddings really storing concepts for all the languages? Say would you pick words in different languages with similar meanings, will the they naturally get similar attention weights? In other words, do multiple languages including very low resource languages learn to naturally align together to the semantic embeddings during multilingual training? I am a bit doubtful especially for the low resource languages.
(4) It seems that the language specific transformation does not always help. Is it because there is not enough data to learn this matrix well?
(5) During multilingual training, how you balance the number of examples for low and high resource languages?

---

> ### Author Response · Authors · 2018-11-16
> **Response to the reviewer**
>
> We thank the reviewer for the comments and insightful questions. Here are some response for the questions:
>
>
> 1) The reviewer is correct that SDE is mainly tailored for multilingual training of related languages with reasonably overlapping character vocabularies.
> a) We choose to focus on related language because multilingual training is most effective when the data from different languages have similar probability distribution. In Table 2 of Gu et al (2018) paper, Korean, the most distant language from other related languages, has the lowest BLEU score with multilingual training, as well as the least improvement even after adding the universal encoder. In fact, it is not clear from their experiments that whether the multilingual training can even outperform training with Korean data only. In Table 2 of Neubig & Hu 2018, they also found that bilingual training of two highly related languages is comparable to or even better than training with all languages.
>
> b) We can still use SDE even if two languages don’t overlap strongly in their character vocabularies. In this case, although the character n-gram embeddings cannot be shared directly between languages, the latent embedding space may still be shared between languages. SDE might behave more like the encoder in Gu et al (2018), but it still has the advantages that (1) it does not need to use word pieces and (2) it can capture morphology etc. within each language through the character n-gram embedding.
>
> c) For languages that do not have natural word boundary, such as Chinese, it is standard to first perform word segmentation and then encode each word with its character embedding using SDE. This might be helpful for co-training of Chinese and Japanese since both languages share part of the character vocabulary.
>
>
>
> 2) Size and training of latent embedding space: The latent embedding matrix is trained jointly with the whole model. We set the number of latent word embedding to 10,000 based on the performance on development set for bel-rus dataset. We started some new experiments varying the latent embedding size of 5000, 10,000, and 15,000 for azetur and belrus dataset, and here are the results on the test set: for azetur, the test BLEU scores are 12.43, 11.66, 10.65 respectively, while the sub-sep baseline is 10.42; for belrus, the test BLEU scores are 18.74, 19.03, 17.76 respectively, while the sub-sep baseline is 17.14. Therefore, SDE out-performs the sub-sep baseline with all three different latent embedding sizes. Moreover, SDE with a relatively small latent embedding size can usually achieve good performance.
>
>
>
>
> 3) Functionality of latent embedding space: The reviewer raised a very interesting question on the kind of information that the latent embedding space stores. While in general neural models are difficult to interpret concretely (even attention in NMT models does not perfectly correspond to word alignments), we can still hypothesize about the functionality of each part of the model especially from the ablation studies. For example, in Table 4 we show that removing the latent word embedding can harm the performance of SDE. Moreover, in Appendix A4, we also added a visualization of the KL divergence of the attention over the latent embedding space between word pairs from two related languages. We found that  similar words from two related languages tend to have similar attention distribution over the latent embedding space.
>
>
>
> 4) Data size and language specific transform: We think that the reviewer proposed an interesting hypothesis that data size might relate to the effectiveness of the language specific transform. Moreover, we think that language specific transform might be more helpful for divergent languages. In Figure 3 of our paper, we find that bel-rus language pair, which benefits the most from language specific transform, has the most divergent vocabularies.
>
>
>
> 5) Balance of HRL and LRL: For all our experiments, we just used the training data as it is and did not balance the training size of HRL and LRL data. In fact, Neubig & Hu 2018 found that balanced sampling does not have significant effect on multilingual training.

---

### Author Response · Authors · 2018-11-16
**Some revisions to the paper**

We thank all the reviewers for your careful comments and great suggestions. Based on your comments and some of our further experiments, we just uploaded a new version of the paper with the following updates:

1) We fixed some typos, made the terminology consistent, and clarified the paper based on the comments.

2) We added two visualization sections in the appendix. One about attention over latent embedding space, and the other about word vectors.

3) We updated the sub-sep baseline result (row 3 in Table 3 and Figure 4). We were looking at the results and found that for the sub-sep baseline only, we used a different subword vocabulary for one of the three random seed runs. After the update, the baseline of bel decreased, while slk improved, aze and glg stay about the same. In general, the updated number doesn’t change any of our conclusions in the paper; SDE still outperforms the strong baseline on all four languages, bringing gains of up to 2.5 BLEU (for the bel-rus dataset).

---

### Meta-Review · Area_Chair1 · 2018-12-14
**well-executed solid work**

**Confidence:** 5
**Recommendation:** Accept (Poster)

**Metareview:**

although some may find the proposed approach as incremental over e.g. gu et al. (2018) and kiela et al. (2018), i believe the authors' clear motivation, formulation, experimentation and analysis are solid enough to warrant the presentation at the conference. the relative simplicity and successful empirical result show that the proposed approach could be one of the standard toolkits in deep learning for multilingual processing.


J Gu, H Hassan, J Devlin, VOK Li. Universal Neural Machine Translation for Extremely Low Resource Languages. NAACL 2018.
D Kiela, C Wang, K Cho. Context-Attentive Embeddings for Improved Sentence Representations. EMNLP 2018.